# Healthcare Professionals’ Perceptions, Barriers, and Facilitators towards Adopting Computerised Clinical Decision Support Systems in Antimicrobial Stewardship in Jordanian Hospitals

**DOI:** 10.3390/healthcare11060836

**Published:** 2023-03-13

**Authors:** Fares Albahar, Rana K. Abu-Farha, Osama Y. Alshogran, Hamza Alhamad, Chris E. Curtis, John F. Marriott

**Affiliations:** 1Department of Clinical Pharmacy, Faculty of Pharmacy, Zarqa University, P.O. Box 2000, Zarqa 13110, Jordan; 2Department of Clinical Pharmacy and Therapeutics, Faculty of Pharmacy, Applied Science Private University, P.O. Box 541350, Amman 11937, Jordan; 3Department of Clinical Pharmacy, Faculty of Pharmacy, Jordan University of Science and Technology, P.O. Box 3030, Irbid 22110, Jordan; 4Department of Pharmacy, College of Medical & Dental Sciences, University of Birmingham, Birmingham B15 2TT, UK

**Keywords:** antimicrobial stewardship, computerized clinical decision support systems, healthcare professionals, tertiary hospitals

## Abstract

Understanding healthcare professionals’ perceptions towards a computerised decision support system (CDSS) may provide a platform for the determinants of the successful adoption and implementation of CDSS. This cross-sectional study examined healthcare professionals’ perceptions, barriers, and facilitators to adopting a CDSS for antibiotic prescribing in Jordanian hospitals. This study was conducted among healthcare professionals in Jordan’s two tertiary and teaching hospitals over four weeks (June–July 2021). Data were collected in a paper-based format from senior and junior prescribers and non-prescribers (*n* = 254) who agreed to complete a questionnaire. The majority (*n* = 184, 72.4%) were aware that electronic prescribing and electronic health record systems could be used specifically to facilitate antibiotic use and prescribing. The essential facilitator made CDSS available in a portable format (*n* = 224, 88.2%). While insufficient training to use CDSS was the most significant barrier (*n* = 175, 68.9%). The female providers showed significantly lower awareness (*p* = 0.006), and the nurses showed significantly higher awareness (*p* = 0.041) about using electronic prescribing and electronic health record systems. This study examined healthcare professionals’ perceptions of adopting CDSS in antimicrobial stewardship (AMS) and shed light on the perceived barriers and facilitators to adopting CDSS in AMS, reducing antibiotic resistance, and improving patient safety. Furthermore, results would provide a framework for other hospital settings concerned with implementing CDSS in AMS and inform policy decision-makers to react by implementing the CDSS system in Jordan and globally. Future studies should concentrate on establishing policies and guidelines and a framework to examine the adoption of the CDSS for AMS.

## 1. Introduction

The prevalence of multidrug resistance has increased alarmingly globally [1]. Moreover, the inappropriate use of antimicrobial agents has been correlated with antimicrobial resistance (AMR) [2,3]. The World Health Organisation (WHO) released a recent feasible toolkit concerning the use of antimicrobial stewardship (AMS) in healthcare settings in low- and middle-income countries to optimize the use of antibiotics and contain the problem of AMR [4]. The AMS can be referred to as a coordinated intervention designed to enhance the appropriate use of antibiotics by promoting the selection of optimal antimicrobial drug regimens, doses, duration of therapy, and routes of administration to achieve the best desired clinical outcomes with low toxicity and minimum antibiotic resistance [5].

In Jordan, the minister of health launched a four-year action plan aligned with the WHO global action plan to maintain the efficiency of existing antibiotics through AMS [6]. Increasing antibiotic resistance at an alarming rate is further fueled by the widespread misuse of antibiotics [7,8]. In Jordan, people store their unused medication, including antibiotics, for future use [9]. This would enhance the risk of self-medication with antibiotics [3,9,10]. Jordan has a high prevalence of self-medication with antibiotics, recently reported at 40% [3,10]. Additionally, 87.8% of all deaths in Jordan are estimated to be secondary to infectious diseases [3]. The national plan was developed considering the importance of healthcare information technology in combating AMR [6]. 

Many information technology systems have been developed to aid clinicians in decision-making [11,12]. One such system is the computerised clinical decision support system (CDSS). The CDSS provides real-time, evidence-based decision support at the point of care about the choice of antimicrobial agents in selected infections [12,13]. CDSS is mostly defined as a computer-based system intended to support clinical decision-making in everyday patient care by presenting to the healthcare worker an integrated summary of clinically relevant patient information [14]. The emergence of automated CDSS is facilitated by the introduction of electronic health records (EHRs) and computerized provider order entry systems (CPOE) [14]. However, the CDSS is less commonly used in clinical practice despite its effectiveness in reducing AMR [12,13,15]. The impact of CDSS has been evaluated in many different clinical settings [12,13,16,17,18]. Some studies showed positive impacts, such as improved patient care processes, health care costs, physician workflows, and guideline adherence [12,13,19,20].

In contrast, others showed a negative effect and failed to achieve their intended outcomes [18,21,22]. CDSS studies fail to report consideration of the non-expert, end-user workflow. They have a narrow focus, such as antimicrobial selection, and use proxy outcome measures. Negative perceptions among healthcare professionals towards CDSS could affect the acceptance of such systems. Understanding the perceptions and attitudes of healthcare professionals towards CDSS may provide a platform for the determinants of the successful adoption and implementation of CDSS. Therefore, it is interesting to study healthcare professionals’ perceptions of CDSS [21,23] and their role in CDSS adoption [24,25]. In addition, this study examines potential barriers and facilitators that hinder or enable CDSS use in clinical practice and AMS. Highlighting these factors would provide a framework for successful CDSS implementation and use.

The success of the implementation and performance of CDSS depends on the technical characteristics of the software, the clinical aspects of the task at hand, and the physician’s expertise with the CDSS [14]. Next to these, a substantial human factor remains, and acceptance of the CDSS is essential. Many interruptive medication alerts are, e.g., simply ignored by the operator [13,14]. In addition, the problem of alert fatigue is a well-established downside of interruptive messaging in CDSS. However, different aspects of the successful implementation of CDSS devices have been explored, mostly in narrow contexts for well-defined and delineated clinical problems. Little evidence is available on which factors should be taken into account to maximize uptake by clinicians when incorporating CDSS into general EHRs/CPOEs [14,16]. Most EHRs/CPOEs available on the market today are designed from an administrative and informatics perspective. They rarely consider the specific requirements of clinical tasks. Most systems do not take into account local conditions and culture, and most offer general solutions to general problems rather than specific solutions to the actual problems the clinicians and their patients are facing. As a consequence, they produce unrealistic, inapt, or plainly unsuitable advice for the local setting [12,16]. Therefore, there is a huge gap between what healthcare workers have to put into the system to make it work, mainly administrative information, and what they get out in terms of improved care for their patients.

After conducting a literature review on health professionals’ perceived facilitators and barriers to CDSS implementation, three main limitations of previous studies were identified. First, the studies primarily focused on technical and usability issues, while they tended to overlook the social, cultural, and contextual factors potentially influencing their implementation [26,27]. Moxey and colleagues [28] suggested that the variability in CDSS uptake may be attributable to the technical aspects of the technology itself. Second, most studies evaluated the perceptions of frontline clinicians but did not address the perceptions of different organizational roles (e.g., hospital administrators, chiefs, or non-physician staff) that are key to establishing the overall mission and vision of the healthcare institution in addition to shaping the expected behavior and standards of its personnel. For example, organizational leadership that supports technological innovation may encourage and reward the use of CDSSs to improve patient care. Third, the studies often addressed contexts in which CDSSs had already been introduced; these studies did not account for the perceived facilitators and barriers existing before CDSS introduction or for the evolution of perceptions throughout the technology’s various stages of uptake. This study aims to identify potential barriers and facilitators to the uptake of CDSS in AMS to curtail the inappropriate use of antibiotics and contain AMR.

## 2. Materials and Methods

### 2.1. Compliance with Ethical Standards

Ethical approval to conduct this study was obtained from the ethics committees at Zarqa University (3/3/2018–2019), Jordan University Hospital (80/2019/23), and King Abdullah University Hospital (49/128/2019). In addition, an informed consent form was collected from all participants before participation in the study, ensuring voluntary participation and that the participants could withdraw at any stage with their answers treated confidentially.

### 2.2. Hospital Setting and Participants

This prospective cross-sectional study was conducted in two tertiary teaching hospitals, the University of Jordan and King Abdullah University Hospital. The University of Jordan Hospital is a large (600-bed) academic centre located in Amman (Middle), Jordan, and King Abdullah University Hospital (750-bed) is another large academic centre located in Irbid (North), Jordan. Both King Abdullah University Hospital and the University of Jordan Hospital have infectious diseases departments that apply AMS principles, which curtail the use of broad-spectrum antimicrobials within the hospital. Two categories of healthcare professionals were invited to participate: medical and allied healthcare professionals. In addition, senior and junior doctors from different specialties were asked to participate in the questionnaire, as were other healthcare professionals, including pharmacists, microbiology experts, and infection control experts. Therefore, the sample included both prescribers and non-prescribers and was less discretional.

### 2.3. Questionnaire Development and Data Collection

The complete questionnaire has been adopted from Zaidi et al. [29], as shown in Appendix A. A pool of ten items to measure barriers and nine items to measure facilitators were initially drafted into a questionnaire tool. Two clinical pharmacists and one infectious diseases consultant reviewed the drafted questionnaire items. These items were then reviewed by three research experts familiar with the study design. The research experts commented on the wording, clarity, and comprehensiveness of the questionnaire items, and whether each item was relevant to the study’s aims and objectives. The research experts’ feedback and comments were reviewed by the authors and used to refine the questionnaire. After completing the piloting process, the final questionnaire version was developed.

The final version of the questionnaire included 38 items divided into four domains. The first domain collected participants’ demographic data such as age, gender, specialty, and experience in a specialist role. The second domain collected information about the perceptions of healthcare professionals toward AMS. The third domain collected information about the awareness of using CDSS and the perceived benefits. The last domain collected information on the perceived barriers and facilitators towards using CDSS. The questionnaire was piloted in the local region, especially with two clinical pharmacists and one infectious diseases consultant, in March 2021. Two clinical pharmacists distributed the questionnaire in a paper-based format for senior and junior prescribers and non-prescribers at the University of Jordan Hospital and King Abdullah University Hospital. Healthcare professionals were informed that participation would be voluntary, information collected would be anonymous, and the questionnaire would take 10–15 min to finish. The questionnaire delivery lasted four weeks, started on 15 June 2021, and was closed on 14 July 2021. A reminder was sent six weeks later.

Before beginning the study, participants were given information outlining the purpose of the study and participant rights and consented to participate. Participants will be notified that their involvement is voluntary, can be withdrawn at any time, and that confidentiality is protected through anonymizing all collected data.

### 2.4. Statistical Analyses

The statistical package for social science (SPSS®) version 29 (SPSS® Inc., Chicago, IL, USA) was used for data analysis. The mean ± SD and frequency (percentages) were used for continuous and categorical variables. The chi-square test was used to evaluate the difference between healthcare providers (clinical pharmacists, nurses, and physicians) in their perception of AMS. Univariate and multivariate logistic regression was employed to screen for factors affecting participants’ awareness of electronic prescribing and electronic health record systems in AMS. Variables that were significant on a single predictor level (*p*-value < 0.25) using univariate logistic regression analysis were contained in the logistic regression analysis model. In the logistic regression analysis, variables independently associated with awareness about the use of electronic prescribing and the electronic health record system in AMS were identified. Statistical significance was considered at *p*-value < 0.05.

## 3. Results

A total of 254 healthcare providers agreed to participate in this study and completed the survey, with a response rate of 45.8% (254 out of 550). The majority of participants (*n* = 193, 76.0%) were aged between 20 and 30 years old, and 59.1% (*n* = 150) were female. About 61% (*n* = 154, 60.6%) of the participants were physicians, while the remaining were clinical pharmacists (*n* = 60, 23.6%) and nurses (*n* = 40, 15.7%). Study Demographics are presented in Table 1.

Participants were asked about their awareness of the use of electronic prescribing and electronic health record systems in general and in AMS (Table 2). The majority (*n* = 220, 86.6%) reported knowing about the presence of electronic prescribing and electronic health record systems at their hospitals. In addition, around three-quarters (*n =* 184, 72.4%) were aware that such systems could be used to facilitate antibiotic use prescribing. However, a lower percentage of the respondents (*n =* 161, 63.4%) were aware that those systems could provide a clinical decision-support function to support evidence-based practice.

All participating healthcare providers responded to five statements to express their perception of AMS (Table 3). First, healthcare providers showed a positive perception of AMS, where 88.2% of respondents (*n* = 224) agreed/strongly agreed that AMS programs might improve patient care. Similarly, 89.7% of respondents (*n* = 228) believed those programs might reduce the problem of AMR. In addition, more than 90% of respondents agreed/strongly that stewardship programs should be incorporated at a hospital level and that healthcare providers should be provided with adequate training on antimicrobial use. On the other hand, only 52.4% of the respondents (*n* = 133) believed that the antimicrobial prescribing at their hospital is already as good as possible, given many initiatives related to AMS launched at their hospitals. When comparing clinical pharmacists, nurses, and physicians, nurses showed the worst perception towards AMS-related statements, except for the first statement, “Antimicrobial prescribing at your hospital is already as good as it can be,” where they scored the highest percentage of agreement (65.0 for nurses versus 57.1 for physicians and 31.7% for clinical pharmacists).

Participants were also asked about their perceived benefits of using electronic prescribing and electronic health record systems in AMS. Results showed that respondents believed that those systems might reduce the expenditure on antibiotics (*n* = 212, 83.4%), improve the safety of antibiotic use (*n =* 210, 82.7%), and may improve the ability to deliver AMS (*n =* 205, 80.8%). More details are included in Figure 1. 

Regarding the barriers against the use of electronic prescribing and electronic health record systems in AMS (Figure 2), results demonstrate that the most important barrier was the insufficient training to use the systems (*n* = 175, 68.9%), followed by the lack of access to reliable technical support (*n* = 173, 68.1%). The least important barrier was the limitation of medical autonomy (*n* = 111, 43.7%). 

On the other hand, Figure 3 illustrates the perceived facilitators of AMS electronic prescribing and electronic health record systems. Results showed that the most important facilitator was making the system available in a portable format such as a mobile device or personal digital assistant (*n* = 224, 88.2%), followed by linking radiology and laboratory results to the system (*n* = 220, 86.6% for both). More details are included in Figure 3. 

Finally, logistic regression (Table 4) showed that female healthcare providers showed significantly lower awareness about using electronic prescribing and electronic health record systems in AMS compared to males (*p* = 0.006). Moreover, nurses showed significantly higher awareness about using those systems in AMS than clinical pharmacists and physicians (*p* = 0.041).

## 4. Discussion

Healthcare providers should have adequate knowledge, awareness, acceptability, and understanding of AMS to implement the program successfully in their hospital settings. CDSSs are integral to implementing the AMS. Therefore, CDSSs will likely become an integral part of clinical practice to improve patient care and safety continually. Well-established clinical workflows and EHRs are important requisites to successfully introducing and using CDSSs in clinical settings. Users should also be provided with sufficient training, education, and support. In addition to developing technical suggestions on CDSS design and implementation, understanding perceived barriers and facilitators to CDSSs is important to maximize the technology’s usage and its potential to impact patient outcomes.

This is a two-centre study (both large tertiary and teaching hospitals) from the middle and northern areas of Jordan in which 254 healthcare professionals provided their insights about their positive perceptions toward AMS, demonstrated by their agreement that its use would improve patient outcomes and curtail AMR. This study aimed to examine the potential barriers and facilitators that hinder or enable CDSS use in clinical practice and AMS based on the healthcare professionals’ views. Therefore, understanding perceived barriers and facilitators to CDSSs is vital to maximise the technology’s adoption and uptake and potentially impact patient outcomes. The study survey adopted was refined after considering various contexts, from two healthcare centres where the healthcare professionals would be unfamiliar with the technology to those in the mature stages of its implementation. 

Consequently, the results from this study are expected to help guide the development of strategies and recommendations essential to introducing and integrating CDSS into wider national healthcare settings, including the two hospital centres.

This study showed that healthcare professionals had positive awareness and perceptions towards electronic prescribing and electronic health record systems, explained by their understanding that AMS use would improve patient outcomes and limit AMR. Moreover, healthcare professionals perceived that electronic prescribing is beneficial and would reduce the high cost of prescribing antibiotics (i.e., reduce the expenditure of antibiotics), improve the efficacy and safety of antibiotic use, and may improve the ability to deliver AMS to optimise the rational use of antibiotics. In addition, many systematic reviews and studies demonstrated the impact of adopting the CDSS on antibiotics management and AMS [18,19,30,31,32,33,34,35,36,37].

The lack of appropriate training to use electronic prescribing and electronic health record systems and the lack of access to reliable technical support were the most perceived barriers to CDSS adoption among healthcare professionals. In addition, more than half of the healthcare professionals had a low level of awareness and were unfamiliar with using electronic prescribing and electronic health record systems. Furthermore, given that healthcare professionals have busy work schedules, they do not have enough time to learn how to use the system. Therefore, adequate training should be encouraged for novice and experienced healthcare professionals to use the CDSS system and improve workflow effectively.

Technical support should be provided to the healthcare professionals in each hospital ward to avoid difficulties in using the systems and maximise the effective use of CDSS. Previous studies reported similar results [29,37,38,39,40,41]. For example, one recent cross-sectional study from Australia that evaluated the impact of CDSS adoption on antibiotics management reported that the lack of appropriate training and technical support was an essential barrier to CDSS adoption [37]. In addition, the significance of training and technical support for CDSS adoption was evident in previous studies [29,42,43,44].

Making the system in an easily portable format (i.e., such as a mobile device or a personal digital assistant) was the most perceived facilitator for adopting CDSS by healthcare professionals. This is expected to enable healthcare professionals to make a decision regarding prescribing and monitoring antibiotics from remote areas without the need to be in the ward or hospital to prescribe and monitor antibiotics, thus making work more flexible. Moreover, lab and radiology results linked to the CDSS are facilitators for adopting CDSS. As a result, healthcare professionals will not be required to check different databases to confirm the diagnosis and adjust treatment based on laboratory results. This will make the work schedule more flexible and efficient. Similar studies reported the same results [29,39,41]. For example, one cross-sectional study conducted in a tertiary care university hospital in Melbourne, Australia, reported making the CDSS in an easily portable format and linking lab and radiology results as a facilitator to the adoption of the CDSS [29]. This was also evident in a recent systematic review [41] and a previous study [39].

The logistic regression results about the factors affecting participants’ awareness about the use of electronic prescribing and electronic health record systems in antimicrobial stewardship showed that female healthcare providers had significantly lower awareness about using electronic prescribing and electronic health record systems. In contrast, the nurses showed significantly higher awareness about using those systems in stewardship programs. Gender as a factor affecting participants’ awareness about the use of electronic prescribing and electronic health record systems was studied in the literature [45,46,47,48], with some studies showing no difference [47] and others showing higher awareness among females [46,48], which contrasts with the results from our study. However, our results were consistent with a recent World Health Organisation report, which reported that females are generally less likely to be involved in such activities and skills in developing countries [49]. Males and females are supposed to have equal technological exposure, knowledge, and awareness; however, this will be further explored in future studies.

The response rate was less than optimal, similar to other response rates in the literature [37,50]; this may affect the representativeness of the data obtained and may be a limitation of this study. Moreover, the majority of healthcare providers range in age from 20–30 years old, with only seven above 40 years, indicating some bias and a lack of generalizability, as those in the younger category will prefer electronic use; however, the decision-makers are usually older. This is considered another limitation of this study.

Using the CDSS was optional to minimise the likelihood that healthcare professionals would be influenced by hospital policy to use the CDSS system. However, this would affect their responses, especially for those who do not routinely use the electronic prescribing system, and thus impact the study findings; this would be another limitation of this study. Nevertheless, it is vital to note that the study attracted healthcare professionals with various degrees of system usage. Therefore, the sample seems adequate to address the study’s aims.

This study has several strengths. First, the sample size of 254 healthcare providers (despite a low response rate = 46%) from two large tertiary teaching hospitals is considered satisfactory and would add to the representativeness of the results drawn from this study. Second, the CDSS system was designed by developers independent of the end-user healthcare professionals. So, the healthcare professionals were not involved in designing and implementing the CDSS. This is expected to reduce the potential for investigator bias. While the CDSS system is increasingly gaining popularity in implementing hospital guidelines for prescribing antibiotics, significant barriers to its adoption exist. To implement CDSS systems successfully, the developer needs to understand the barriers to their adoption. The present study measures healthcare professionals’ perceptions of using the CDSS system in two tertiary care settings in Jordan’s middle and northern areas. Both the study setting and the study participants represent a metropolitan area. Therefore, the findings from the study could apply to other healthcare settings interested in the launch and implementation of CDSS systems. While the study investigators are independent of the developer and implementer of the CDSS system at the study hospitals, the present study results have been available to them to improve the implementation and deployment strategies.

## 5. Conclusions

This study examined healthcare professionals’ perceptions towards adopting CDSS for antibiotic prescribing in Jordan’s two tertiary and teaching hospitals. Findings from this study would help get more insight into the perceived barriers and facilitators to adopting and implementing CDSS, which would help reduce antibiotic resistance and improve patient safety. Moreover, results would help provide a platform for other healthcare settings interested in implementing CDSS in AMS and inform policy decision-makers to react by implementing the CDSS system in Jordan and globally. Future studies should focus on establishing guidelines and a policy framework to examine the adoption of the CDSS for AMS.

## Figures and Tables

**Figure 1 healthcare-11-00836-f001:**
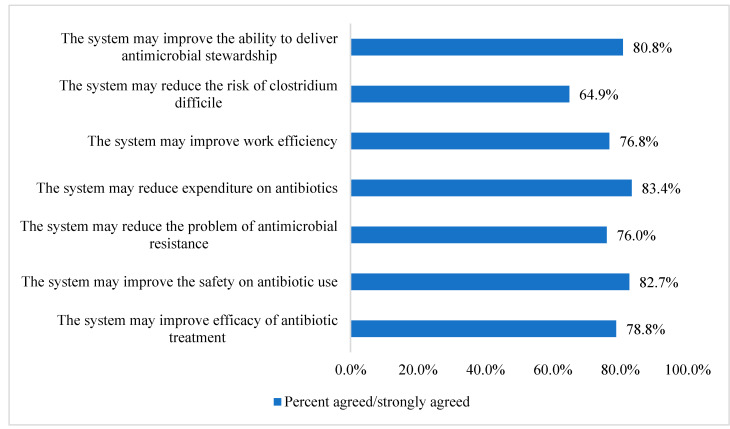
Participants perceived the benefits of using electronic prescribing and electronic health record systems in antimicrobial stewardship (*n* = 254).

**Figure 2 healthcare-11-00836-f002:**
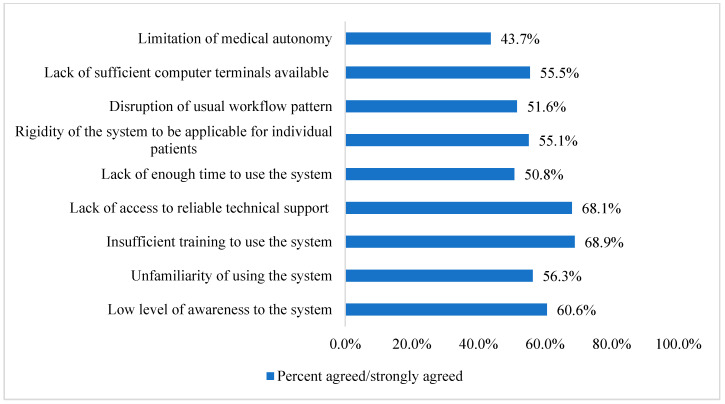
Participants perceived barriers to using electronic prescribing and electronic health record systems in antimicrobial stewardship (*n* = 254).

**Figure 3 healthcare-11-00836-f003:**
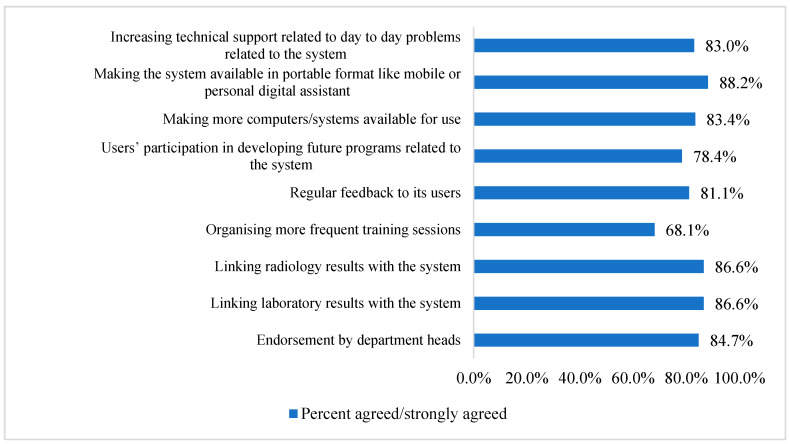
Participants perceived facilitators to use electronic prescribing and electronic health record systems in antimicrobial stewardship (*n* = 254).

**Table 1 healthcare-11-00836-t001:** Socio-demographic characteristics of the study sample (*n =* 254).

Parameter	*n* (%)
Age (years)	
20–30	193 (76.0)
31–40	54 (21.3)
41–50	5 (2.0)
51–60	0 (0.0)
>61	2 (0.8)
Gender	
Males	104 (40.9)
Females	150 (59.1)
Speciality	
Clinical pharmacist	60 (23.6)
Physicians	154 (60.6)
Nurse	40 (15.7)

**Table 2 healthcare-11-00836-t002:** Participants’ awareness about the use of electronic prescribing and electronic health record systems in antimicrobial stewardship (*n =* 254).

Statements	Strongly Agreed/Agreed
Have you previously used electronic prescribing and electronic health record systems?	
Yes	220 (86.6)
No/Not sure	34 (13.4)
Are you aware that electronic prescribing and electronic health record systems can be used to facilitate antibiotic prescribing?	
Yes	184 (72.4)
No/Not sure	70 (27.6)
Are you aware that electronic prescribing and electronic health record system is capable of providing clinical decision support function in order to support evidence-based practice?	
Yes	161 (63.4)
No/Not sure	93 (36.6)

**Table 3 healthcare-11-00836-t003:** Perception of participants towards antimicrobial stewardship (*n =* 254).

	Strongly Agreed/Agreed *n* (%)				
Statements	Total	Clinical Pharmacists *n =* 60	Nurses *n =* 40	Physicians *n =* 154	*p*-Value #
Antimicrobial prescribing at your hospital is already as good as it can be	133 (52.4)	19 (31.7)	26 (65.0)	88 (57.1)	0.001 *
Antimicrobial stewardship programs may improve patient care.	224 (88.2)	59 (98.3)	31 (77.5)	134 (87.0)	0.005 *
Antimicrobial stewardship should be incorporated at a hospital level	229 (90.2)	56 (93.3)	27 (67.5)	146 (94.8)	<0.001 *
Antimicrobial stewardship programs may reduce the problem of antimicrobial resistance.	228 (89.7)	57 (95.0)	31 (77.5)	140 (90.9)	0.014 *
Adequate training should be provided to healthcare professionals on antimicrobial use	235 (92.5)	58 (96.7)	30 (75.0)	147 (95.5)	<0.001 *

# Using chi-square test, * significant at 0.05 significance level.

**Table 4 healthcare-11-00836-t004:** Assessment of factors affecting participants’ awareness about the use of electronic prescribing and electronic health record systems in antimicrobial stewardship (*n* = 254).

Parameter	Awareness [0: No, 1: Yes]			
	OR	*p*-Value #	OR	*p*-Value $
Age (years) 20–30 years >31 years	Reference 3.135	0.005 ^	2.130	0.079
Gender Male Female	Reference 0.394	0.003 ^	0.402	0.006 *
Speciality Physicians Clinical pharmacists Nurses	Reference 1.322 4.327	0.410 0.008 ^	1.636 3.249	0.172 0.041 *

# Using simple logistic regression, $ using multiple logistic regression, ^ eligible for entry in multiple logistic regression, * significant at 0.05 significance level.

## Data Availability

The data presented in this study are available on request from the corresponding author.

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
