# Peer review of "Healthcare Professionals’ Perceptions, Barriers, and Facilitators towards Adopting Computerised Clinical Decision Support Systems in Antimicrobial Stewardship in Jordanian Hospitals"

_healthcare, 2023, doi:10.3390/healthcare11060836_

Round 1
Reviewer 1 Report
The authors here present a new method to assess the barriers that physicians deal with when facing with antimicrobial prescriptions and infectious diseases, under the scope of the AMS project.
The work is well written and very comprehensive. I consider authors did a good job. However, I would like if authors could discuss a little bit a few things, for example:
- They found significance differences among age groups and genders. I understand the age gap using new tech, while why do authors consider that could be based on the acceptance differences seen between female and male? Can you please discuss a little bit about it?
- In the abstract, authors say (line 21-22) that "most were aware that electronic prescribing could be used to...", what does exactly mean? The rest were not informed and did not understand what it was the data collected for?
- In line 52 authors say "others showed a negative effect and failes to achieve their intended outcomes". Could you please provide the most relevant examples for that, here?
- Line 100-102: I found it confusing If it lasted 4 weeks how it was a reminder sent after 4 weeks?
- Table 3: There is a high percentage of responses that disagree with the first statement (antimicrobial prescrbing at your hospital is already as ood as it can be). Could you please discuss about it?
- Lines 168-171, line 236: Why do you consider may be the reason for those differences?
- Have you found differences tendencies between hospitals or all the data were collected together?
Minor:
Line 28: AMS, describe the acronym the first time use.
Line 65: as shown in appendix A
Line 87-88 rephrase to: "These comments were used to develop the final vwrsion of the questionnaire, which included 38 items divided into four domains."
Line 94-95: repetitive, already said.
Line 99: will be voluntaria
Line 147: More detailes are included in Figure 1.
Figure 1: Scientific names in italics
Line 164: More detailes are included in Figure 3.
Figure 3: The statemens next to the bars are with " ... ", please include the full sentences.
Author Response
Dear Reviewer 1,
Thank you for your valuable comments,
Reviewer 1
Comment 1:
The authors here present a new method to assess the barriers that physicians deal with when facing with antimicrobial prescriptions and infectious diseases, under the scope of the AMS project. The work is well written and very comprehensive. I consider authors did a good job. However, I would like if authors could discuss a little bit a few things, for example:
Response to comment 1:
Thank you for your valuable comments; for sure. We will address all your comments.
Comment 2:
They found significance differences among age groups and genders. I understand the age gap using new tech, while why do authors consider that could be based on the acceptance differences seen between female and male? Can you please discuss a little bit about it?
Response to comment 2:
Thank you for your valuable comments; corrected as advised.
Comment 3:
- In the abstract, authors say (line 21-22) that "most were aware that electronic prescribing could be used to...", what does exactly mean? The rest were not informed and did not understand what it was the data collected for?
Response to comment 3:
Thank you for your valuable comment; the sample size included junior doctors and other allied healthcare professionals who use the electronic prescribing system for their own specific practice setting, which is not solely about antibiotic prescribing. Also, the use of electronic prescribing was optional. This would be why they did not recognize the importance of the electronic prescribing system on antibiotic prescribing. However, to address your valuable point, changes (tacked) were made, and this is now addressed as a limitation to this study.
Comment 4:
- In line 52 authors say "others showed a negative effect and failes to achieve their intended outcomes". Could you please provide the most relevant examples for that, here?
Response to comment 4:
Thank you for your valuable comment, most relevant examples were provided now.
Comment 5:
- Line 100-102: I found it confusing If it lasted 4 weeks how it was a reminder sent after 4 weeks?
Response to comment 5: Thank you for your valuable comment, change as advised.
Comment 6:
- Table 3: There is a high percentage of responses that disagree with the first statement (antimicrobial prescrbing at your hospital is already as ood as it can be). Could you please discuss about it?
Response to comment 6: Thank you for your valuable comment, there is a high percentage of responses who agree
Comment 7:
- Lines 168-171, line 236: Why do you consider may be the reason for those differences?
Response to comment 7: Thank you for your valuable comment, corrected as advise
Comment 8:
- Have you found differences tendencies between hospitals or all the data were collected together?
Response to comment 8: Thank you for your valuable comment; we don't find any differences in tendencies between hospitals
Minor:
Comment 9:
Line 28: AMS, describe the acronym the first time use.
Response to comment 9: Antimicrobial stewardship was added as an acronym AMS
Comment 10:
Line 65: as shown in appendix A
Response to comment 10:Thank you for your valuable comment; corrected as advised.
Comment 11:
Line 87-88 rephrase to: "These comments were used to develop the final version of the questionnaire, which included 38 items divided into four domains."
Response to comment 11: Thank you for your valuable comment; the whole sentence was re-phrased.
Comment 12:
Line 94-95: repetitive, already said.
Response to comment 12: Thank you for your valuable comment; repetitive words were removed
Comment 13:
Line 99: will be voluntaria
Response to comment 13: Thank you for your valuable comment; ‘will be voluntarily’ was added
Comment 14:
Line 147: More detailes are included in Figure 1.
Response to comment 14: Thank you for your valuable comment;the statement was corrected as requested to “More details are included in Figure 1”
Comment 15:
Figure 1: Scientific names in italics
Response to comment 15: Thank you for your valuable comment; corrected as advised.
Comment 16:
Line 164: More detailes are included in Figure 3.
Response to comment 16: Thank you for your valuable comment;the statement was corrected as requested to “More details are included in Figure 3”
Comment 17:
Figure 3: The statemens next to the bars are with " ... ", please include the full sentences.
Response to comment 17: Thank you for your valuable comment;Figure 3 format was corrected so; all the sentences are complete.
Reviewer 2 Report
Suggestions for improvement
1. Add details pertaining to study duration, sample size, how the sample size arrived, method of sampling, validation of the tool, pilot-testing, any operational definitions
2. Add strengths of the study and limitations
3. Make conclusion specific to the study
Author Response
Dear Reviewer 2,
Thank you for your valuable comments,
Reviewer 2
Comment 1:
Add details pertaining to study duration, sample size, how the sample size arrived, method of sampling, validation of the tool, pilot-testing, any operational definitions
Response to comment 1: Thank you for your valuable comment;all details were added
Comment 2:
Add strengths of the study and limitations
Response to comment 2: Thank you for your valuable comment; corrected as advised.
Comment 3:
Make a conclusion specific to the study
Response to comment 3: Thank you for your valuable comment; corrected as advised.
Reviewer 3 Report
"Healthcare professionals’ perceptions, barriers, and facilitators towards adopting computerized clinical decision support systems in antimicrobial stewardship in Jordanian hospitals" This study was performed in one of the developing countries. The rational and content are weak. This idea would be interested couple of years ago, not in 2023. A higher steps in AMS is required; comparison between different tools; assess the outcome, develop a plan etc. The readers will not know the level of AMS and electronic resources in Jordan from this manuscript. More data are needed in the introduction and discussion. Even, in developing countries, some hospitals use electronic and others use the traditional ways etc. I have the following major comments:
Abstract: overall, is very weak
1. Line 17-19: There are unnecessary repeated words
2. Line 21,22: The authors mentioned that the sample size is 254, then they stated that "majority (n= 84, 72.4%" which is wrong simple statistics; it might be 184?
3. Line 23, 24: wrong grammar
4.AMS abbreviation was not defined in the abstract!
5. No clear clear conclusion
6. Arrange the keywords alphabatically.
7. a keyword with 5 words is too much
Introduction: overall, weak
1. Many used references are old and there are many new related references
2. The sentences need more links between them; unpleasant reading and flow
3. Reference 17 need date and location
4. The references style is not consistent with the Journal style
5. The limitations of current knowledge and why this study was done is not obvious. The readers would be interested to know to which level is AMS is currently in Jordan. The authors have to mention other studies done in Jordan related to the perception of the healthcare provides and may be the community toward the use of the electronic databases in the general and also for AMS. Accordingly, there is no strong rational to perform this study.
Methods:
1. "Ethical approval to conduct this study was obtained from the ethics committee at Zarqa University (3/3/2018-2019), Jordan University Hospital (80/2019/23), and King Abdullah University Hospital (49/128/2019), as shown in (Appendix A)." However, in Appendix A only the university approval is included; either to add them all or nothing, depending on the Journal requirement, or at least to move (Appendix A) to the first institution.
2. What you mean with "non-medical healthcare professionals."? add details
3. Line 101 "A reminder was sent four weeks later." this sentence is unclear?
4. SPSS v22 is old.
Results:
1. Line 116-119: Please, be consistent in the decimal points, also throught the manuscript.
2. In the text, you mentioned pharmacists and in the table clinical pharmacists, be clear and consistent.
3. Table 3 needs further statistics; does the difference is significant in term of chi square? or convert the answers into numbers and show the average etc.
4. Figures 1 to 3 are basic and descriptive; most appropriate in a local magazine.
5. The studied factors/parameters in Table 4 are not enough and lack important implications. The two groups according to the age are greatly different making the insignificant difference expected.
Discussion:
Low response rate
1. All of the discussion part is introduced as one paragraph making it hard to read.
2. There is a lack of clear comparison with other studies including those from developing countries.
3. The authors did not mention the major limitations: The majority are 20-30 years of age. Only 7 above 40 years, indicating some bias and lack of generalizability; those young category, of course, will prefer the electronic use, however, the decision makers are usually older. There are no clear questions if the prefer to use the electronic resources over the traditional methods.
Author Response
Dear Reviewer 3,
Thank you for your valuable comments,
Reviewer 3:
"Healthcare professionals’ perceptions, barriers, and facilitators towards adopting computerized clinical decision support systems in antimicrobial stewardship in Jordanian hospitals" This study was performed in one of the developing countries. The rational and content are weak. This idea would be interested couple of years ago, not in 2023. A higher steps in AMS is required; comparison between different tools; assess the outcome, develop a plan etc. The readers will not know the level of AMS and electronic resources in Jordan from this manuscript. More data are needed in the introduction and discussion. Even, in developing countries, some hospitals use electronic and others use the traditional ways etc. I have the following major comments:
Response: Thank you for your valuable comments; we will revise the manuscript as per comments.
Comment 1:
Abstract: overall, is very weak
Response to comment 1: Thank you for your valuable comments; Abstract is revised as advised.
Comment 2:
Line 17-19: There are unnecessary repeated words
Response to comment 2: Thank you for your valuable comments; The unnecessary repeated words were removed from the abstract
Comment 3:
Line 21,22: The authors mentioned that the sample size is 254, then they stated that "majority (n= 84, 72.4%" which is wrong simple statistics; it might be 184?
Response to comment 3: Thank you for your valuable comments; Yes, it is a typo, it was corrected to 184.
Comment 4:
Line 23, 24: wrong grammar
Response to comment 4: Thank you for your valuable comments; grammar corrected
Comment 5:
AMS abbreviation was not defined in the abstract!
Response to comment 5: Thank you for your valuable comments; AMS abbreviation was defined in the abstract
Comment 6:
No clear clear conclusion
Response to comment 6: Thank you for your valuable comments; The conclusion is now revised.
Comment 7:
Arrange the keywords alphabatically.
Response to comment 7: Thank you for your valuable comments; all keywords were arranged alphabatically.
Comment 8:
a keyword with 5 words is too much
Response to comment 8: Thank you for your valuable comments; one keyword has been removed
Comment 9:
Introduction: overall, weak
Response to comment 9: Thank you for your valuable comments; the Introduction is now revised as per your comments.
Comment 10:
Many used references are old and there are many new related references
Response to comment 10: Thank you for your valuable comments; new references are added now.
Comment 11:
The sentences need more links between them; unpleasant reading and flow
Response to comment 11: Thank you for your valuable comments; the flow and the linkage between sentences were improved
Comment 12:
Reference 17 need date and location
Response to comment 12:Refernce is deleted as it is old conference paper.
Comment 13:
The references style is not consistent with the Journal style
Response to comment 13: Thank you for your valuable comments;now corrected as advised
Comment 14:
The limitations of current knowledge and why this study was done is not obvious. The readers would be interested to know to which level is AMS is currently in Jordan. The authors have to mention other studies done in Jordan related to the perception of the healthcare provides and may be the community toward the use of the electronic databases in the general and also for AMS. Accordingly, there is no strong rational to perform this study.
Response to comment 14: Thank you for your valuable comments; few studies related to AMS in Jordan were added to the introduction
Methods:
Comment 15:
- "Ethical approval to conduct this study was obtained from the ethics committee at Zarqa University (3/3/2018-2019), Jordan University Hospital (80/2019/23), and King Abdullah University Hospital (49/128/2019), as shown in (Appendix A)." However, in Appendix A only the university approval is included; either to add them all or nothing, depending on the Journal requirement, or at least to move (Appendix A) to the first institution.
Response to comment 15: Thank you for your valuable comments; added as advised
Comment 16:
What you mean with "non-medical healthcare professionals."? add details
Response to comment 16: Thank you for your valuable comments; changed to allied health care professionals
Comment 17:
Line 101 "A reminder was sent four weeks later." this sentence is unclear?
Response to comment 17: Thank you for your valuable comments; A reminder was sent 6 weeks later to collect as many responses as possible
Comment 18:
SPSS v22 is old.
Response to comment 18: Thank you for your comment; This is the version we can access.
Results:
Comment 19:
Line 116-119: Please, be consistent in the decimal points, also throught the manuscript.
Response to comment 19: Thank you for your comment; The entire result section was revised to ensure that all numbers were reported with one decimal point (except for the p-values)
Comment 20:
In the text, you mentioned pharmacists and in the table clinical pharmacists, be clear and consistent.
Response to comment 20: Thank you for your comment; The word “pharmacists” in the text and table 4 was changed to “clinical pharmacists”
Comment 21:
Table 3 needs further statistics; does the difference is significant in term of chi square? or convert the answers into numbers and show the average etc.
Response to comment 21: Thank you for your comment; Table 3 was revised, where we have compared between the different specialties (nurses, clinical pharmacists and physicians) in term of their perception towards AMS using chi-square test.
Comment 22:
Figures 1 to 3 are basic and descriptive; most appropriate in a local magazine.
Response to comment 22: we respect your offensive comment; however, some descriptive statistics are needed to describe the general perception of the study sample benefits, barriers, and facilitators of the AMS.
Comment 23:
The studied factors/parameters in Table 4 are not enough and lack important implications. The two groups according to the age are greatly different making the insignificant difference expected.
Response to comment 23: Thank you for your comment; Unfortunately, we did not collect that actual age for healthcare providers, and as we see in table 1, 75% of the healthcare providers were between 20-30 years. That’s why to be able to conduct regression analysis we combined the other groups. This may be a reason why age showed a non-significant results. This was added to the limitation section.
Discussion:
Comment 24: Low response rate
Response to comment 24: Thank you for your comment; added as a limitation of this study.
Comment 25:
All of the discussion part is introduced as one paragraph making it hard to read.
Response to comment 25: Thank you for your comment;revised as advised.
Comment 26:
There is a lack of clear comparison with other studies including those from developing countries.
Response to comment 26: Thank you for your comment; corrected as advised
Comment 27:
The authors did not mention the major limitations: The majority are 20-30 years of age. Only 7 above 40 years, indicating some bias and lack of generalizability; those young category, of course, will prefer the electronic use, however, the decision makers are usually older. There are no clear questions if the prefer to use the electronic resources over the traditional methods.
Response to comment 26: Thank you for your comment; your comment is revised and addressed as a limitation to this study.
Round 2
Reviewer 3 Report
The authors addressed most of the points and the manuscripts is improved.